# Scientific multi-agent reinforcement learning for wall-models of turbulent flows

H. Jane Bae [1,2✉] & Petros Koumoutsakos [1,3✉]

The predictive capabilities of turbulent flow simulations, critical for aerodynamic design and weather prediction, hinge on the choice of turbulence models. The abundance of data from experiments and simulations and the advent of machine learning have provided a boost to turbulence modeling efforts. However, simulations of turbulent flows remain hindered by the inability of heuristics and supervised learning to model the near-wall dynamics. We address this challenge by introducing scientific multi-agent reinforcement learning (SciMARL) for the discovery of wall models for large-eddy simulations (LES). In SciMARL, discretization points act also as cooperating agents that learn to supply the LES closure model. The agents self-learn using limited data and generalize to extreme Reynolds numbers and previously unseen geometries. The present simulations reduce by several orders of magnitude the computational cost over fully-resolved simulations while reproducing key flow quantities. We believe that SciMARL creates unprecedented capabilities for the simulation of turbulent flows.

[1] School of Engineering and Applied Sciences, Harvard University, 29 Oxford Street, Cambridge, MA 02138, USA. [2] Graduate Aerospace Laboratories, California Institute of Technology, 1200 E. California Boulevard, Pasadena, CA 91125, USA. [3] Computational Science and Engineering Laboratory, ETH Zurich, Clausiusstrasse 33, Zurich CH-8092, Switzerland. ✉email: jbae@caltech.edu; petros@seas.harvard.edu

Simulations of wall-bounded turbulent flows have become a key element in the design cycle of wind farms[1] and aircraft[2] and the major factor in the predictive capabilities of simulations of atmospheric flows[3]. Due to the high Reynolds numbers associated with these flows, direct numerical simulations (DNS), where all scales of motion are resolved, are not attainable with current computing capabilities. LES aims to reduce the necessary grid requirements by resolving only the energy-containing eddies and modeling the smaller scale motions. However, this requirement is still hard to meet in the near-wall region, as the stress-producing eddies become progressively smaller, scaling linearly in size with the distance to the wall. Several studies[4–6] have estimated that the number of grid points necessary for wall-resolved LES scales as $\mathcal{O}(Re^{13/7})$, where $Re$ is the characteristic Reynolds number of the flow. This number of computational elements is orders of magnitude smaller than that required for DNS, yet it remains prohibitive. In turn, modeling the near-wall flow such that only the large-scale motions in the outer region of the boundary layer are resolved, the grid-point requirement for wall-modeled LES (WMLES) scale at most as $\mathcal{O}(Re)$. With WMLES, certification by analysis—prediction of the aerodynamic quantities of interest for engineering applications by numerical simulations alone—may soon be a reality. Certification by analysis is expected to narrow the number of wind tunnel experiments, reducing both the turnover time and cost of the design cycle.

Several strategies for modeling the near-wall region have been explored[7–10]. The taxonomy of WMLES approaches can be broadly categorized as Hybrid LES/RANS methods and wall-flux modeling. Hybrid LES/RANS and its variants[8] combine Reynolds-averaged Navier–Stokes (RANS) equations close to the wall and LES in the outer layer, with the interface between RANS and LES domains enforced implicitly through the change in the turbulence model. In wall-flux modeling, the usual no-slip and thermal wall boundary conditions are replaced with stress and heat flux boundary conditions provided by the wall model. Examples of well-known approaches involve computing the wall stress using either the law of the wall[11–13] or the RANS equations[14–20]. Models account for nonlinear advection and pressure gradient by solving the unsteady three-dimensional RANS equations[15,17] or accounting only for the wall-normal diffusion reducing the computational requirements to the solution of a system of ordinary differential equations[19,20].

The main impediments of the above-mentioned models are that they rely on RANS parametrization that requires the use of a priori empirical coefficients calibrated for a particular flow state (usually fully developed turbulence in equilibrium over a flat plate). Such wall models do not function as intended in real-world applications, where various flow states coexist (e.g., separated flows, flow over roughness, predicting transition, etc.)[7]. The use of RANS parametrization for wall modeling was challenged with a dynamic wall model that is free of a priori specified coefficients at a negligible additional cost[21,22]. The two approaches were formulated by requiring consistency between the filtered velocity field at the wall and a differential filter kernel.

Dynamic wall models provide encouraging results, but they also face significant challenges. They are robust for changes in Reynolds number and grid resolution but sensitive to numerical methods employed in the flow solver and the choice of the subgrid-scale (SGS) model. This is attributed to the dominance of numerical errors close to the wall that in turn affect the evaluation of the necessary wall-model constants[23]. Furthermore, the methodology has only been exploited specifically for structured, incompressible flow solvers, with limited applicability for compressible flows or complex geometries.

The essential requirements for a successful dynamic wall model are that it (i) accommodates diverse flow solvers and SGS models and (ii) generalizes beyond their calibration flow fields. Recent advances in machine learning and data science aim to address these issues and complement the existing turbulence modeling approaches. To date, most efforts have focused on the application of supervised learning to SGS modeling[24–30] and wall modeling[31–33]. However, despite the demonstrated promise, these methods encounter difficulties in generalizing beyond the distributions of the training data. In supervised learning, the parameters of the neural network are commonly derived by minimizing the model prediction error, which is often based on single-step target values to limit computational challenge. Therefore, it is necessary to differentiate between a priori and a posteriori testing. The first measures the accuracy of the supervised learning model in predicting the target values on a database of reference simulations, typically obtained via DNS. A posteriori testing is performed after training, by integrating in time the Navier–Stokes equations along with trained supervised learning closure and comparing the obtained statistical quantities to that of DNS or other reference data. Due to the single-step cost function, the resultant neural network model is not trained to compensate for the systematic discrepancies between DNS and LES (or WMLES) and the compounding errors. The issue of ill-conditioning of data-driven SGS models has been exposed by studies that perform a posteriori testing[27,34–36]. Wall models are more sensitive than SGS models[22], and we expect the compounding of errors to play a more detrimental role in WMLES.

Here, we propose SciMARL for the development of wall models in LES. Reinforcement learning (RL) identifies optimal strategies for agents that perform actions, contingent on their information about the environment, and measures their performance via scalar reward functions. In this work, the agents correspond with the computational elements and their actions compensate both for the closure terms and errors associated with the numerics of the flow solver. RL is a semi-supervised learning framework with foundations on dynamic programming[37] and a broad range of applications ranging from robotics[38,39], games[40,41], and more recently flow control[39,42–45]. We note that SciMARL has been used in fluid mechanics only recently for the development of SGS models in LES of homogeneous turbulent flow[46].

In the case of WMLES, the performance of the SciMARL can be measured by comparing the statistical properties of the simulation to those of reference data such as the wall-shear stress. SciMARL is a semi-supervised learning algorithm that requires information about the flow formulated in terms of a reward rather than detailed spatiotemporal data as in the case of supervised learning. In the case of wall modeling, SciMARL does not rely on a priori knowledge of the log-law coefficients but rather aims to discover active closure policies according to patterns in the flow physics captured by the filtered equations. The respective wall models are robust with respect to the numerical discretizations, as these errors are taken into consideration in the training process. Furthermore, the model discovery method can be readily extended to complex geometries and different flow configurations, such as flow over rough surfaces and stratified and compressible boundary layers.

## Results

**Multi-agent reinforcement learning for wall modeling**. In RL, the agent interacts with its environment by sampling its states ($s$), performing actions ($a$), and receiving rewards ($r$). At each time step, the agent performs the action and the system is advanced in time before the agent can observe its new state, receive a scalar reward, and choose a new action. The agent infers a policy $\pi(s, a)$

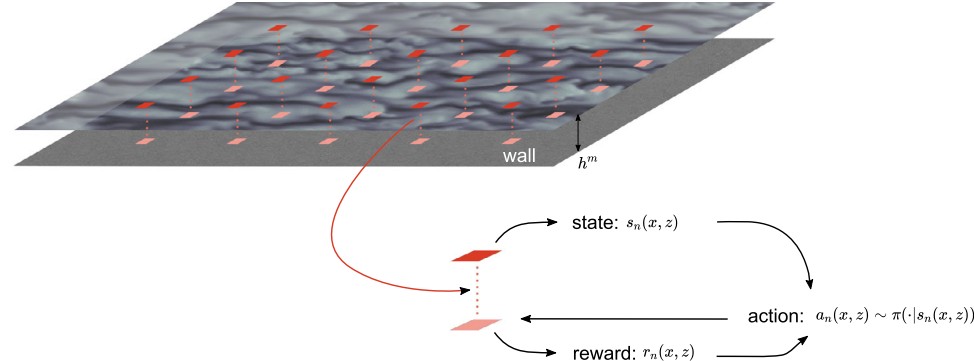

**Fig. 1 Diagram of the SciMARL setup.** Agents are distributed evenly along the wall, with each agent obtaining state information wall-normal height $h^m$ away from the wall, computing the reward at the wall and supplying into the policy $\pi$ to obtain actions $a$ for the next time step.

through its repeated interactions with the environment to maximize its long-term rewards. The optimal policy $\pi^*(s, a)$ is found by maximizing the expected utility, which is given by the expected cumulative reward. Throughout the paper, $x$, $y$, and $z$ denote the streamwise, wall-normal, and spanwise directions, respectively. The corresponding velocity components are $u$, $v$, and $w$. RL agents are distributed evenly on each channel wall with each agent located at $(x, z)$ receiving local states $s_n(x, z)$ and rewards $r_n(x, z)$ and providing local actions $a_n(x, z)$ at each time step $t_n$. A single policy is maintained and updated by the multiple agents present in the domain (Fig. 1).

In order for the RL to be universally applicable for a wide range of flow parameters, the states are nondimensionalized using viscosity $v$ and the modeled instantaneous friction velocity

$$u_\tau^m(x, z, t) = \left( \frac{\tau_w^m(x, z, t)}{\rho} \right)^{1/2}, \tag{1}$$

where $\tau_w^m$ is the modeled wall-shear stress and $\rho$ is the density. These quantities are only dependent on the output of the wall model and can be obtained without any prior knowledge of the flow. This non-dimensionalization is noted by the superscript $*$ and is distinct from the one by the true friction velocity $u_\tau$, noted by the superscript $+$, which will be used for the assessment of model performance. The goal of the wall model is to predict the correct wall-shear stress $\tau_w$, and thus the $u_\tau$, which will allow for good predictions of quantities such as the mean velocity profile and turbulence intensities[47].

**Velocity-based wall model.** We first train the model to adapt to the variation of the velocity with the wall-normal height, which has a universal behavior in the log region. We set as states $s_n(x, z)$ the instantaneous velocity $u^*(x, h^m, z, t_n)$, the wall-normal derivative $\partial u^*/\partial y^*(x, h^m, z, t_n)$, and the wall-normal location $y^* = (h^m)^*$ of the sampling point. Agents act to adjust the wall-shear stress through a multiplication factor $a_n(x, z) \in [0.9, 1.1]$ such that $\tau_w^m(x, z, t_{n+1}) = a_n(x, z)\tau_w^m(x, z, t_n)$. This choice does not require the model to produce the exact wall-shear stress (which is dependent on Reynolds number), but rather proposes an action that adjusts the wall-shear stress. The reward (see "Methods" for definition) is also incremental and proportional to the improvement in the prediction of the wall-shear stress compared to the one obtained in the previous time step. The agent behavior is rendered stable by providing additional reward if the predicted wall-shear stress is within 1% of the true value.

**Log-law-based wall model.** The second model is based on the existence of a logarithmic (log) layer in the near-wall region of turbulent flows, present in all flows with an inner–outer scale

separation[48]. In the log layer, the velocity profile is expressed as:

$$u^+ = \frac{1}{\kappa}\log y^+ + B, \tag{2}$$

where $\kappa$ is the von Karman constant and $B$ is the intercept constant. The exact value of $\kappa$ and $B$ depends on the flow configurations and wall roughness; however, for the current study, we use values attributed to a canonical smooth zero-pressure-gradient boundary layer. The states for the second model are the local instantaneous coefficients for the log law $\kappa^m$ and $B^m$, computed from the instantaneous velocity, velocity gradient, and wall-normal location information. We emphasize that this model does not take as input the a priori known values of $\kappa$ and $B$ from the log law, but rather derived quantities from the instantaneous flow. This has an advantage over the first model in that the values do not depend on the value of $y^*$ and thus the model can learn the log-law behavior for $y^*$ outside the range of values it trained on. This allows the model to be extended to higher Reynolds numbers or coarser grids more readily. The actions and rewards are the same as the first model.

**State-action map.** We inquire into the learned models by examining the state-action map conditioned to positive rewards for the channel flow at friction Reynolds number $Re_\tau = 2000, 4200, 8000$. As seen in Fig. 2a, the velocity-based wall model (VWM) has distinguished states $(y^*, u^*)$ with distinct actions corresponding to positive rewards. The model is able to up/down-shift the wall-shear stress based on whether the $(y^*, u^*)$ pair is located above or below the log-law profile. The model initially does not have any prior knowledge of the log-law coefficients, yet it is able to learn to adjust the wall-shear stress through the RL process. However, because the model is trained on a limited range of $(h^m)^+$ in the training set, the extrapolation of this behavior to much larger values of $(h^m)^+$ may be challenging. This can be alleviated by refining the grid in the wall-normal direction with $N_y \sim Re^{4-6}$.

The log-law-based wall model (LLWM) similarly has distinct states with different actions corresponding to positive rewards (see Fig. 2b). The main mechanism for controlling the wall-shear stress is similar to the VWM, with the wall-shear stress being up/down-shifted based on whether the point corresponding to the slope and intercept of $1/\kappa^m$ and $B^m$ are under/over-predicting the log law. Depending on the wall-normal location of $h^m$, the classification of whether the point is above or below the log law may vary, especially for points farther away from the origin. However, the majority of the states are centered around the true value of $1/\kappa$ and $B$, and the mechanism will work as expected.

**Testing: turbulent channel flow.** We examine the model predictions on turbulent channel flow for Reynolds numbers in the range from 5200 to $10^6$ (Fig. 3). In the case of VWM, we expect

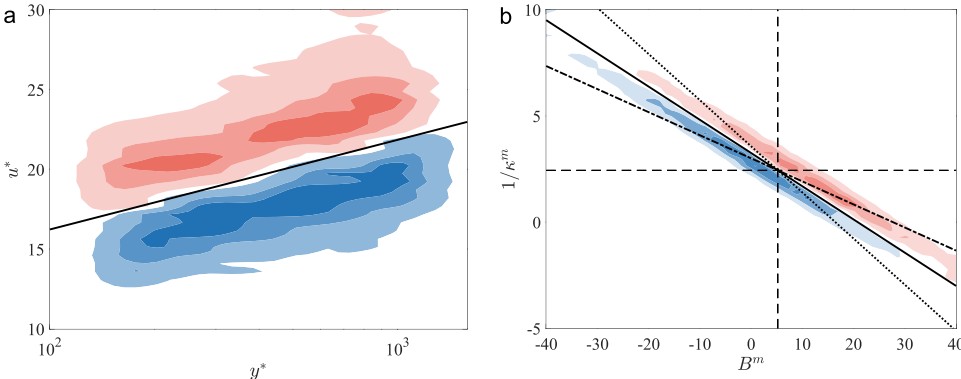

**Fig. 2 State-action map of VWM and LLWM. a** Probability density function of states $u^*$ for given $y^*$ for VWM conditioned to events with $r > 0.1$ and $a < 0.95$ (blue) or $a > 1.05$ (red). Contour levels are 30, 50, 70% of maximum value. Line indicates the log law $u^+ = 1/\kappa \log y^+ + B$ with $\kappa = 0.41$, $B = 5.2$. **b** Joint probability density functions of states $1/\kappa^m$ and $B^m$ for LLWM conditioned to events with $r > 0.1$ and $a < 0.95$ (blue) or $a > 1.05$ (red). Contour levels are 30, 50, 70% of maximum value. Dashed lines indicate $\kappa = 0.41$, $B = 5.2$; the solid, dotted, and dot-dashed lines are $(1/\kappa^m - 1/\kappa) \log(y^+) + (B^m - B) = 0$ for $(h^m)^+ = 500$, 100 and $10^4$, respectively.

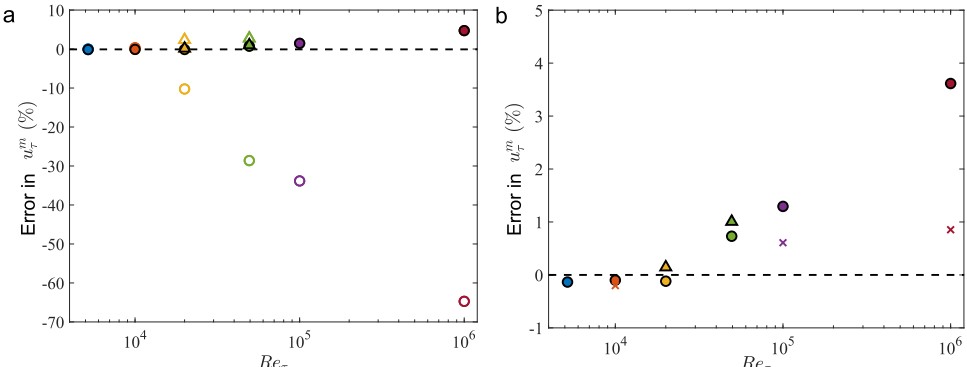

**Fig. 3 Errors in the friction velocities. a** Error in time-averaged wall-shear stress obtained from the VWM (empty) and LLWM (filled) for various Reynolds numbers. Circles indicate the standard grid with $\Delta_y = 0.05$ and triangles indicate refined cases. **b** Zoomed in the version of (**a**) for LLWM with error in EQMW (crosses) for three Reynolds numbers.

that as long as $(h^m)^+$ is within the range observed during the training process ($150 < (h^m)^+ < 1200$), the model will perform as expected. Cases at $Re_\tau = 2 \times 10^4$ and $5 \times 10^4$ produce high errors as the $(h^m)^+$ is not within the trained range. Once the values of $(h^m)^+$ are adjusted to be within the range by refining the grid, the errors decrease significantly. This entails refining the grid for higher Reynolds numbers to allow the first grid point off the wall to be within the trained range of $(h^m)^+$. In the case of LLWM, we observe that the prediction error in the friction velocity is less than 4% while the mean velocity profiles are well-aligned with the log law regardless of the value of $(h^m)^+$. The error increases with Reynolds number, most likely due to the high variation of the streamwise wall-normal gradient with increasing Reynolds number as well as the departure of $(h^m)^+$ from the trained range of values. Still, the results are comparable to the results obtained from the widely used equilibrium wall model (EQWM) up to $Re_\tau \approx 10^5$, which uses an empirical coefficient tuned for this particular flow configuration. This range of Reynolds numbers is sufficient for various external aerodynamic and geophysical flows. The predicted mean velocity profiles for both models are shown in Fig. 4.

**Testing: spatially evolving turbulent boundary layer.** The predictive performance of the LLWM is assessed in a zero-pressure-gradient flat-plate turbulent boundary layer. The simulation

ranges from $Re_\theta = 1000$–7000, where $Re_\theta$ is the Reynolds number based on the momentum thickness.

The modeled skin friction coefficient $C_f^m = \tau_w^m/(\rho U_\infty^2/2)$ for the full simulation domain is comparable to the $C_f$ from the empirical values[49] (Fig. 5a). This shows that the model is capable of adapting to variations of wall-shear stress in the streamwise direction, even when it was only trained on a channel-flow simulation.

**Distribution of wall-shear stress.** A growing body of studies in wall-bounded turbulence has shown that the generation of wall-shear stress fluctuations is directly connected with outer-layer large-scale motions[50,51]. This observation supports the idea that the log-layer flow contains the information necessary to predict not only the mean wall-shear stress but also the fluctuations. However, in deterministic wall models such as the EQWM, the wall-shear stress is perfectly correlated with the velocity at the sampling location[52,53], as opposed to a correlation coefficient of 0.3 observed in DNS[50]. This can be observed in Figs. 6a and 7, where the wall-shear stress predicted by the EQWM is perfectly correlated with the velocity fluctuations at the sampling location $h_{wm}$. On the other hand, LLWM results in a smaller correlation between the velocity at an off-wall location and the wall-shear stress (Figs. 6b and 7) with a maximum correlation of ~0.3, which matches the expected correlation from DNS.

**Potential of SciMARL wall models.** We demonstrate that the SciMARL wall models perform as well as the RANS-based EQWM,

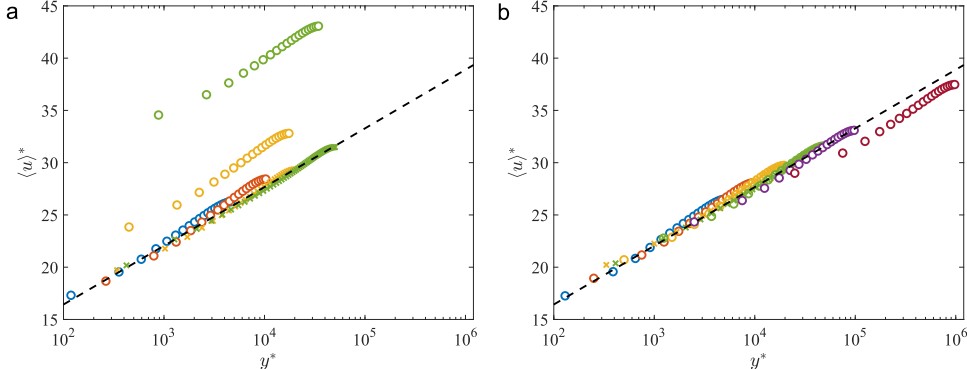

**Fig. 4 Predicted mean velocity profiles for turbulent channel flow.** Mean velocity profiles for the (**a**) VWM cases and (**b**) LLWM cases are shown in Fig. 3. Dashed line is $u^+ = 1/\kappa \log(y^+) + B$ for $\kappa = 0.41$ and $B = 5.2$. The two largest Reynolds number cases for VWM are omitted as the velocity profiles are outside the plotted range.

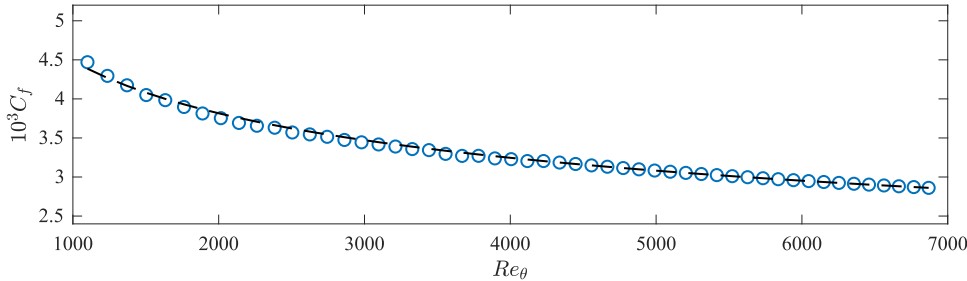

**Fig. 5 Predicted friction coefficients for the turbulent boundary layer.** Friction coefficient $C_f$ as a function of $Re_\theta$. Symbols are LLWM and line is the empirical $C_f$[49].

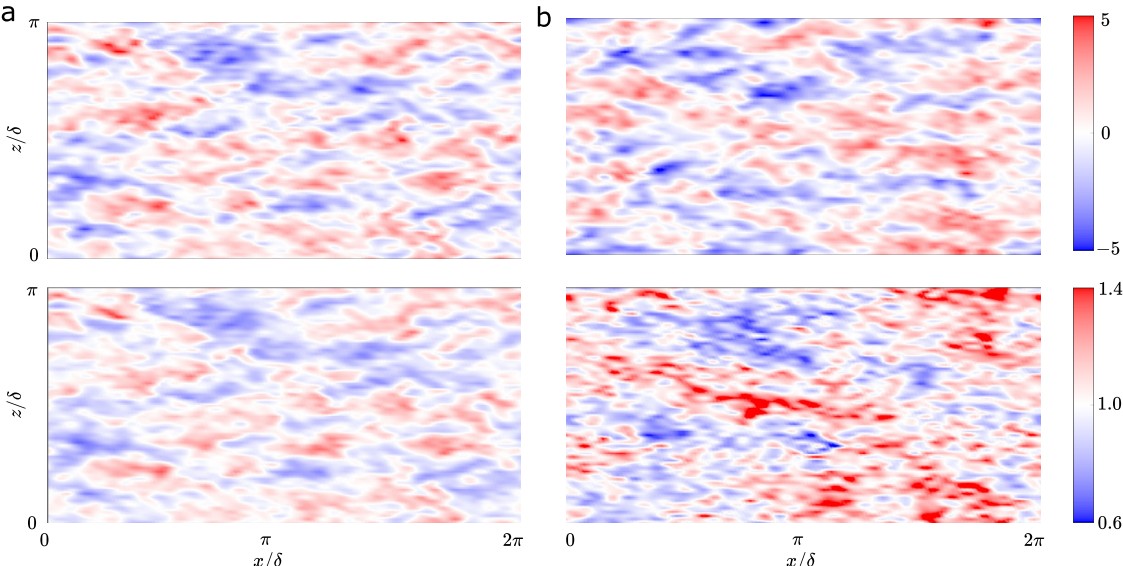

**Fig. 6 Comparison of the instantaneous off-wall streamwise velocity and wall-shear stress.** Instantaneous snapshots of $x$–$z$ plane of streamwise velocity fluctuation $u'^*$ at $h^m$ (top) and $\tau_w^m/\tau_w$ (bottom) for (**a**) EQWM and (**b**) LLWM.

which has been tuned for this particular flow configuration. The SciMARL wall model is able to achieve these results by training on moderate Reynolds number flows with a reward function only based on the mean wall-shear stress rather. Moreover, RL models are trained in-situ with WMLES and do not require any DNS simulation data. This is in contrast to supervised learning methods, where a vast amount of data need to be generated using high-fidelity DNS simulations to proceed with the learning process. For example, in the case of a moderate Reynolds number channel flow

($Re_\tau = 4200$), LLWM can be trained using $O(10^3)$ CPU-hours with less than 1 GB of storage. For supervised learning, generating the DNS data will require $O(10^7)$ CPU-Hours with more than 100 TB storage. DNS databases might be already available for canonical cases such as channel flow, but it would be more difficult to obtain for cases regarding wall roughness or adverse pressure gradients, where wall models will be more useful. The additional overhead for generating data for supervised learning makes it less practical for real-world applications of wall modeling.

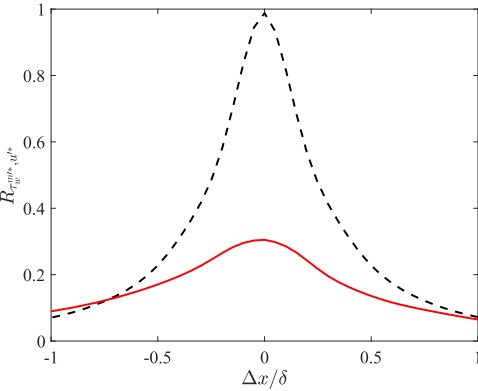

**Fig. 7 Correlation of instantaneous off-wall streamwise velocity and wall-shear stress.** Cross-correlation coefficient between the wall-shear stress $\tau_w^{m\prime*}$ and streamwise velocity $u^{\prime*}$ at sampling location $h^m = 0.1\delta$ for LLWM (red solid line) and EQWM (black dashed line).

The LLWM is easy to extend to complex geometries and flow simulations utilizing different numerical methods or SGS models, as it only takes as states the instantaneous streamwise (or wall-parallel) velocity, its wall-normal gradient, and the distance from the wall. These quantities do not depend heavily on numerics or SGS models, unlike filtered velocities or eddy viscosity values required in the dynamic model[22]. Thus, the model can be used in a wide range of simulations, much like the EQWM, but without prescribed tunable parameters. Furthermore, the RL framework can be extended to various flow configurations by adding an additional dimension to the state vector. Since all flow with an inner–outer scale separation exhibit a log law[48] in the overlap region, the current configuration for wall-model development can be extended to flows exhibiting roughness, stratified flows, compressible flows, among many others. These flows usually have different log-law coefficients $\kappa$ and $B$ that are manually tuned for existing wall models. However, in this work, these values are adjusted automatically using a SciMARL-based model." This gives the LLWM a distinct advantage over existing models. For example, in cases with varying pressure gradients over the simulation domain, traditional methods will have to assign different model parameters for each location containing different pressure gradients. In contrast, the SciMARL model can smoothly transition between various pressure-gradient effects with a single policy trained from various canonical cases when the parameters such as pressure and velocity gradients are included as a state. A similar argument can be applied to simulations with varying levels of stratification or compressible effects within a simulation domain. In addition, the evaluation of the LLWM involves evaluating the weights of the trained neural net, which is an order of magnitude faster than the EQWM that solves an ODE at each time step.

## Discussion

We have introduced a potent method for the automated discovery of closures in simulations of wall-bounded turbulent flows that uses limited data by fusing scientific computing and multi-agent reinforcement learning (SciMARL). In this method, we solve the filtered Navier–Stokes equations using LES and develop a wall model as a control policy enacted by cooperating agents using the recovery of the correct mean wall-shear stress as a reward. SciMARL requires limited data in contrast to supervised learning methods. The training was performed using LES of a turbulent channel flow at moderate Reynolds numbers ($Re_\tau = 2000$, 4200, and 8000). Remarkably, the method generalizes on LES of a turbulent boundary layer and turbulent channel flow at extreme Reynolds numbers.

We examine the robustness of the method by studying two models (VWM and LLWM) with different state spaces. In the VWM, the state space comprises the streamwise velocity and its wall-normal derivatives. This model adjusts the wall-shear stress based on the discrepancy of the velocity profile from the log law. The model captures the mean velocity profile for a wide range of Reynolds numbers when the wall-normal location of the sampling point is within the training set. Alternatively in the LLWM, the state space is based on the instantaneous log-law coefficients. This model generalizes to a broader set of grid resolutions and Reynolds numbers than the VWM. Moreover, despite training in turbulent channel flows we find that the LLWM generalizes to spatially evolving turbulent boundary layer and it recovers the correct skin friction coefficient at a fraction of the cost of high-fidelity simulations.

We note that the LLWM produces correlations between the predicted wall-shear stress and the off-wall velocity profile that are similar to fully resolved flow. This is in contrast to the correlations obtained by the classical RANS models. This implies that the policy of the LLWM replicates the natural mechanisms of wall-shear stress control that can be obtained so far only through highly resolved simulations. Furthermore, as the model only requires instantaneous flow information at one off-wall location, it could be extended to more complex geometries and different numerical methods without additional modifications.

We anticipate that the model can be easily expanded for all wall-bounded flows that exhibit a log law through an inner–outer scale separation[48]. We envision that when SciMARL is trained over a wide range of flows, the model will also acquire experiences for the key flow patterns that are omnipresent in the fundamental physics of flows in complex configurations. This advance will present a paradigm shift in wall-model development for LES in the prediction and control of industrial aerodynamics and environmental flows.

## Methods

**Reinforcement learning**. Learning is performed through the open-source RL library smarties[54]. The library leverages efficiently the computing resources by separating the task of updating the policy parameters from the task of collecting interaction data. The flow simulations are distributed across workers who collect, for each agent, experiences organized into episodes,

$$E_i = \{s_n^{(i)}, r_n^{(i)}, \mu_n^{(i)}, \sigma_n^{(i)}, a_n^{(i)}\}_{n=0,\ldots,N},$$

where $n$ tracks in-episode RL steps, $\mu$ and $\sigma$ are the statistics of the Gaussian policy used to sample $a$, and $t_N$ is the final time step for each episode. When a simulation concludes, the worker sends one episode per agent to the central learning process (master) and receives updated policy parameters. The master stores the episodes to a replay memory (RM), which is sampled to update the policy parameters according to Remember-and-Forget Experience Replay (ReF-ER)[54]. ReF-ER is combined with an off-policy actor-critic algorithm V-RACER which supports continuous state and action spaces.

V-RACER trains a neural network defined by weights $w$ which, given input state $s$, outputs the mean $\mu^w(s)$ and standard deviation $\sigma^w(s)$ of the policy $\pi^w$ and a state-value estimate $v^w(s)$. The statistics $\mu^w$ and $\sigma^w$ are improved through the policy gradient estimator

$$g_\pi(w) = \mathbb{E}\left[g_{\pi,n}(w) \equiv \frac{\pi^w(a_n|s_n)}{\mathcal{P}(a_n|\mu_n, \sigma_n)}(\hat{q}_n - v^w(s_n))\nabla_w \log \pi^w(a_n|s_n)\Big| \{s_n, r_n, \mu_n, \sigma_n, a_n\} \sim \text{RM}\right], \quad (3)$$

where $\mathcal{P}(a_n|\mu_n, \sigma_n)$ is the probability of sampling $a_n$ from $\mathcal{N}(\mu_n, \sigma_n)$, and $\hat{q}_n$ is an estimator of the action value which is computed recursively from a Retrace algorithm[55] as

$$\hat{q}_n = r_{n+1} + \gamma v^w(s_{n+1}) + \gamma \min\left\{1, \frac{\pi^w(a_n|s_n)}{\mathcal{P}(a_n|\mu_n, \sigma_n)}\right\}(\hat{q}_{n+1} - v^w(s_{n+1})), \quad (4)$$

where $\gamma = 0.995$ is the discount factor for rewards into the future. Retrace is also

used to derive the gradients for the state-value estimate

$$g_v(w) = \mathbb{E}\left[ g_{v,n}(w) \equiv \min\left\{1, \frac{\pi_w(a_n|s_n)}{\mathcal{P}(a_n|\mu_n,\sigma_n)}\right\}(\hat{q}_n - v^w(s_n)) \right|$$
$$\{s_n, r_n, \mu_n, \sigma_n, a_n\} \sim \text{RM} \Bigg]. \tag{5}$$

The expectations in Eqs. (3) and (5) are approximated by Monte Carlo sampling $B$ observations from RM.

Due to the use of experience replay, V-RACER and similar algorithms become unstable if the policy diverges from the distribution of experiences in the RM. We circumvent this issue by using an importance weight $\rho_t$ to classify whether an experience is "near-policy" or "far-policy" and clip the gradients computed from far-policy samples to zero[54]. In ReF-ER, the gradients are computed as

$$\hat{g}_n(w) = \begin{cases} \beta g_n(w) - (1-\beta)g_n^D(w), & \text{if } 1/C < \rho_t < C \\ -(1-\beta)g_n^D(w), & \text{otherwise}, \end{cases} \tag{6}$$

where $\rho_t = \pi_w(a_t|s_t)/\mathcal{P}(a_t|\mu_t,\sigma_t)$. Here, $g^D = \nabla_w D_{KL}(\pi_w(\cdot|s_t) \parallel \mathcal{P}(\cdot|s_t))$, where $D_{KL}(P\|Q)$ is the Kullback–Leibler divergence measure between distributions $P$ and $Q$. The coefficient $\beta$ is iteratively updated to keep a constant fraction of samples in the RM within the trust region by

$$\beta \leftarrow \begin{cases} (1-\eta)\beta, & \text{if } r_{RM} > D, \\ \beta + (1-\eta)\beta, & \text{otherwise}, \end{cases} \tag{7}$$

where $r_{RM}$ is the fraction of the RM with importance weights outside the trust region [1/C, C] and $D$ is a parameter.

The most notable hyperparameters used in our description of the MARL setup are the spatial resolution for the interpolation of the actions onto the grid (determined by $\Delta_x^m/\Delta_x$, and $\Delta_z^m/\Delta_z$). The default values $\Delta_x^m/\Delta_x$, and $\Delta_z^m/\Delta_z$ reduce the number of experiences generated per simulation to $O(10^5)$. This value is similar to the number of experiences generated per simulation used for SciMARL of SGS model development[46]. Consistent with previous studies, we found that further reducing the number of agents per simulation reduced the model's adaptability and therefore exhibit slightly lower performance. Because we use conventional reinforcement learning update rules in a multi-agent setting, single parameter updates are imprecise. We found that ReF-ER with hyperparameters $C = 1.5$ and $D = 0.05$ (Eqs. (6) and (7)) stabilizes training. We ran multiple training runs per reward function and whenever we vary the hyperparameters, but we observe consistent training progress regardless of the initial random seed.

Further implementation details of the algorithm can be found in Novati et al.[54].

**Overview of the training setup.** The models are trained on turbulent channel flow simulations of $Re_\tau = u_\tau \delta/\nu \approx 2000, 4200,$ and $8000$, where $\delta$ is the channel half-height with grid resolution $\Delta_{x,y,z} \simeq 0.05\delta$. Each WMLES is initialized for uniformly sampled $Re_\tau \in \{2000, 4200, 8000\}$ with the initial velocity field for the training obtained by superposing white noise sampled from $\mathcal{N}(0, 0.5u_\tau)$ to a previously obtained WMLES flow field at the given $Re_\tau$ that is run for a short period of time to remove numerical artifacts. The initial wall-shear stress is set to overestimate or underestimate the correct wall-shear stress within ± 20%. At each time step of the WMLES, the location $h^m$ is randomly selected between $0.075\delta$ and $0.15\delta$ to train over a smooth range of $(h^m)^+$ within the log layer. The velocity and its wall-normal gradient are then interpolated to the chosen wall-normal location $h_m$ to form the state vector. The agents are located with spacings $\Delta_x^m = 4\Delta_x$ and $\Delta_z^m = 4\Delta_z$. Each iteration of the learning algorithm runs the simulation for $2\delta/u_\tau$, updating the model at every time step.

The policy is parameterized by a neural network with two hidden layers of 128 units each, with soft sign activations and skip connections. The neural network is initialized with small outer weights and bias shifted such that the initial policy is approximately $\mathcal{N}(1, 10^{-4})$[56]. Gradients are computed with Monte Carlo estimates with sample size $B = 512$ from an RM of size $10^6$. The parameters are updated with the Adam algorithm[57] with learning rate $\eta = 10^{-5}$. ReF-ER hyperparameters of $C = 1.5$ and $D = 0.05$ are used to stabilize training. Each training run is advanced for $10^7$ policy gradient steps.

For both VWM and LLWM, the action is given by a multiplication factor $a_n(x,z) \in [0.9, 1.1]$ such that $\tau_w^m(x,z,t_{n+1}) = a_n(x,z)\tau_w^m(x,z,t_n)$. The reward is given by

$$r_n(x,z) = \left( \frac{|\tau_w - \tau_w^m(x,z,t_n)| - |\tau_w - \tau_w^m(x,z,t_{n-1})|}{\tau_w} \right) + \mathbb{1}\left( \frac{|\tau_w - \tau_w^m(x,z,t_n)|}{\tau_w} < 0.01 \right), \tag{8}$$

where $\mathbb{1}$ is an indicator function and $\tau_w$ is the true mean wall-shear stress. This gives a reward that is proportional to the improvement in the prediction of the wall-shear stress compared to the one obtained in the previous time step with an additional reward if the predicted wall-shear stress is within 1% of the true value. The states of the VWM are the instantaneous velocity $u^*(x, h^m, z, t_n)$, the wall-normal derivative $\partial u^*/\partial y^*(x, h^m, z, t_n)$, and the wall-normal location $y^* = (h^m)^*$ of

the sampling point. The states of the LLWM are

$$\frac{1}{\kappa^m}(x,z,t_n) = \left( \frac{\partial u^*}{\partial y^*} y^* \right)(x, h^m, z, t_n), \text{ and } B^m(x,z,t_n)$$
$$= u^*(x, h^m, z, t_n) - \frac{1}{\kappa^m}(x,z,t_n)\log(h^m)^*.$$

**Details of the flow simulation.** We solve the filtered incompressible Navier–Stokes equations in a channel using LES with a staggered second-order finite difference in space[58] with a fractional-step method[59] and a third-order Runge–Kutta time-advancing scheme[60]. The SGS model is given by the anisotropic minimum dissipation (AMD) model[61], which is known to perform well in highly anisotropic grids[62].

For the channel flow, periodic boundary conditions are imposed in the streamwise and spanwise directions, and the no-slip and no-penetration boundary conditions at the top and bottom walls. The modeled wall stress $\tau_w^m$ is applied to the LES domain through the eddy viscosity at the wall[63],

$$\nu_t|_w = \left( \frac{\partial u}{\partial y} \right)\Big|_w^{-1} \frac{\tau_w^m}{\rho} - \nu, \tag{9}$$

where $\nu_t$ is the eddy viscosity and the subscript $w$ indicates values evaluated at the wall. This boundary condition, compared to the more widely used Neumann boundary condition, is better at resolving the so-called log-layer mismatch for WMLES[63]. The channel is driven by a constant pressure gradient for the testing cases. For training, the channel is driven by a constant mass flow rate computed from the mean velocity profile of channel flow. The domain size is given by $L_x = 2\pi\delta$, $L_y = 2\delta$, and $L_z = \pi\delta$, where $\delta$ is the channel half-height.

For the spatially evolving boundary layer, periodic boundary conditions are imposed in the spanwise direction. No-slip and no-penetration boundary condition with viscosity augmentation (Eq. (9)) is used at the wall. In the top plane, we impose $u = U_\infty$ (free-stream velocity), $w = 0$, and $v$ estimated from the known experimental growth of the displacement thickness for the corresponding range of Reynolds numbers[49]. This controls the average streamwise pressure gradient, whose nominal value is set to zero. The turbulent inflow is generated by the recycling scheme[64], in which the velocities from a reference downstream plane, $x_{ref}$, are used to synthesize the incoming turbulence. The reference plane is located well beyond the end of the inflow region to avoid spurious feedback[65,66]. A convective boundary condition is applied at the outlet with convective velocity $U_\infty$[67] with a small correction to enforce global mass conservation[66]. The spanwise direction is periodic.

The code has been validated in previous studies in turbulent channel flows[22,68–70] and flat-plate boundary layers[22,71].

**Testing: channel flow.** The model predictions of VWM and LLWM are tested on turbulent channel flow for Reynolds numbers in the range from 5200 to $10^6$ (see Table 1) and for a time span of $300\delta/u_\tau$ significantly longer than the training period $2\delta/u_\tau$. While only results using $\Delta_x \approx \Delta_z \approx 0.05\delta$ are reported here, using different grid resolutions representative of WMLES also produce similar results.

Note that for LLWM, one of the states, $1/\kappa^m = (\partial u^*/\partial y^*)y^*$, depends on the choice of $y$ with respect to the discrete points of the simulation. For example, if $y$ is located at the midpoint of two computational grid points, a central finite difference can be used to compute the wall-normal derivative $\partial u^*/\partial y^*$. On the other hand, if $y$ is located on the computational grid point, either a left- or right-finite difference is used. In this case, we chose $y$ values that are midpoints of the two computational grid points. Changing the location of $y$ had minor effects on the results, with the wall-shear stress changing ~5% when the location of $y$ was chosen to be on the computational grid point.

**Table 1 List of channel flow test cases and corresponding Reynolds number, wall-normal grid resolution, and matching location $h^m$.**

| $Re_\tau$ | $\Delta_y/\delta$ | $(h^m)^+$ |
|---|---|---|
| 5200 | 0.05 | 520 |
| $10^4$ | 0.05 | 1000 |
| $2 \times 10^4$ | 0.05 | 2000 |
| $2 \times 10^4$ | 0.025 | 1000 |
| $5 \times 10^4$ | 0.05 | 5000 |
| $5 \times 10^4$ | 0.01 | 1000 |
| $10^5$ | 0.05 | $10^4$ |
| $10^6$ | 0.05 | $10^5$ |

For all cases, $\Delta_{x,z}/\delta = 0.05$.

**Testing: spatially evolving turbulent boundary layer**. The predictive performance of LLWM is assessed in a zero-pressure-gradient flat-plate turbulent boundary layer with $Re_\theta$ ranging from 1000 to 7000. This range was chosen so that the results can be compared against relevant DNS[72]. The recycling plane for the inlet boundary condition is set to $x_{ref}/\theta_0 = 890$, where $\theta_0$ is the momentum thickness at the inlet. The length, height and width of the simulated box are $L_x = 3570\theta_0$, $L_y = 100\theta_0$, and $L_z = 200\theta_0$. The streamwise and spanwise resolutions are $\Delta_x/\delta = 0.06$ ($\Delta_x^+ = 128$) and $\Delta_z/\delta = 0.05$ ($\Delta_z^+ = 105$) at $Re_\theta = 6500$. The grid is uniform in the wall-normal direction with $\Delta_y/\delta = 0.03$ ($\Delta_y^+ = 64$) at $Re_\theta = 6500$. The number of wall-normal grid points per boundary layer thickness is chosen to be ~10 at the inlet, which is in line with the channel flow simulations. The sampling point $h^m$ was chosen to be at the third grid point off the wall in the wall-normal direction[16], which places the point in the log region for most of the domain. All computations were run for 50 washout times after transients.

## Data availability

All the data analyzed in this paper were produced with an in-house flow solver and an open-source reinforcement learning software described in the code availability statement. Reference data and the scripts used to produce the data figures is available through GitHub (https://github.com/hjbae/SciMARL_WMLES).

## Code availability

The wall-modeled large-eddy simulations were performed with a in-house flow solver, which is available on demand. The wall models were trained with the reinforcement learning library smarties (https://github.com/cselab/smarties).

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

## Acknowledgements

The authors acknowledge the support of Air Force Office of Scientific Research (AFOSR) Multidisciplinary University Research Initiative (MURI) project: Prediction, Statistical Quantification, and Mitigation of Extreme Events Caused by Exogenous Causes or Intrinsic Instabilities under grant number FA9550-21-1-0058. Computational resources were provided by the Swiss National Supercomputing Centre (CSCS) Project s929.

## Author contributions

H.J.B. jointly conceived the study with P.K., designed and performed experiments, analyzed the data, and wrote the paper; P.K. devised the concept of SciMARL, supervised the project, and edited the manuscript.

## Competing interests

The authors declare no competing interests.
