## [Peer Review File · Nature Communications]

REVIEWER COMMENTS

Reviewer #1 (Remarks to the Author):

Report to the manuscript entitled "Scientific multi-agent reinforcement learning for wall-models of turbulent flows"

The authors developed a multi-agent reinforcement learning (SciMARL) method for the automated discovery of wall models in large-eddy simulations (LES) of wall-bounded turbulent flows. They treat discretization points as cooperating agents that learn to supply the closure model and propose a control policy by cooperating agents using the recovery of the correct mean wall-shear stress as a reward. The paper shows that the SciMARL method requires limited data by using LES of a turbulent channel flow at moderate Reynolds numbers in the training processes. Two models with different state spaces, including a velocity-based wall model (VWM) and a log-law-based wall model (LLWM), are considered. The paper concludes that the proposed method can be generalized for LES of a turbulent boundary layer and turbulent channel flow at extreme Reynolds numbers. The paper may be considered for publication in Nature communication after a major revision and the following issues must be addressed:

1. Please provide more comments about the advantages of SciMARL models, as well as the dependency of SciMARL models on the training data or the predefined theoretical model. In the situation of the velocity-based wall model (VWM), the training processes are dependent on the LES of a turbulent channel flow at moderate Reynolds numbers, and the model cannot be easily generalized to turbulent channel flow at very high Reynolds numbers. The limitation of the proposed VWM is similar to the situations of supervised learning models, which are limited by the scope of training data. In the situation of the log-law-based wall model (LLWM), the training processes are dependent on the well-known theoretical log-law model. For the general complex flows which are not consistent with the theoretical log-law model, the proposed LLWM could fail. For the flows which are consistent with the theoretical log-law model, the original log-law model is enough.
2. Please provide the computational cost for LES of wall-bounded turbulent flows using the new models, and make a comparison with traditional wall models.
3. Please give some comments about the effects of different parameters of SciMARL models on the accuracy of LES.

Reviewer #2 (Remarks to the Author):

This paper identifies an important research question in computational fluid mechanics: developing approximate boundary conditions for Large Eddy Simulations without employing experimental data or direct numerical simulation data. The authors have designed two reinforcement learning approaches to solve this problem. One of these methods (LLWM) is performing quite well. This method assumes the logarithmic law for the velocity profile and determines two unknown coefficients κ and B . The authors have claimed that their approach is a significant advance over the state of the art because rather than assume κ and B , it has learned them through unsupervised learning. However, the manuscript does not state how the truth values of these turbulence coefficients (κ and B) are established and provided to the reinforcement learning agents. If these target values are taken from prior experiments or direct numerical simulations (DNS), then I believe this paper is overstating its significance since the authors acknowledge that there is already a model (EQWM) that uses assumed coefficients to make accurate predictions. On the other hand, I would be very interested to learn if the authors have some novel way to establish the truth values for κ and B (and τ_w for the velocity

based model), which does not require prior information (experiment/DNS). Upon close reading of the manuscript, I have not been able to determine how the truth values of κ and B are determined.

I have given the authors the benefit of the doubt and tried to write this review under the assumption that the authors can modify the manuscript to demonstrate that κ and B are discovered rather than assumed. If this is not the case, the present model, in my opinion, is not any more predictive of the wall stress than the classical EQWM. The principle finding of this paper would then be regarding the correlation of the velocity and wall stress fluctuations. The paper would need to be revised to emphasize that result, which is interesting, though perhaps not suitable for publication in Nature Communications.

Overall the paper was well written, but the organization was a bit unintuitive. Perhaps there could be a section between the "introduction" and "results" sections. Or perhaps that is forbidden by the format requirements of this journal. More specific comments regarding this will follow.

For future revisions, it would be helpful if the manuscript had line numbers.

Comments on the introduction

- "such wall models can only provide limited success" It might be helpful to add references that outline some of the regimes where the RANS models are failing (separated flows, predicting transition, etc.) to help motivate the present work. "Limited success" is fairly vague.
- "SciMARL does not rely on a priori knowledge" the accuracy of this statement depends on how the authors address my comment regarding truth values κ , B , and τ_w .

Comments on the section "Results"

- The "results" section also describes the model. Is this common in this journal?
- Eq. 1, the notation for the expectation under policy π should be explicitly mentioned since presently E_π is not defined. The summation over n is not defined. n is defined in the methods section, but must be defined when it is used in Eq. 1. It would be helpful to note that γ will be specified in the methods section since it was unclear that this was a constant parameter when it was introduced in the results section and not specified.
- A single policy is made by all the agents. This seems like it is only the case in homogeneous flows. In general problems, there will need to be n agents with n policies. In a channel, it seems like only 1 agent is needed. In a boundary layer, one would need more than 1 agent if κ and B are thought to vary with the streamwise coordinate. (However, this variation is likely modest.) Are the authors using additional agents in the span just to improve the statistical convergence?

Comments on the section "Velocity-based wall model"

- It might be helpful to refer the reader to the methodology section since the term "reward" is used without being defined. Especially the bonus to the reward that is designed to promote stability. The reader may not realize that this will be defined later.

Comments on the section "State-action map"

- I don't see where the authors mention which flow they are plotting. I assume it is a channel.
- For figure 2, I think the a simpler explanation will be possible by splitting 2b into 2 panels (2b and 2c) for plots of κ vs y^* and B vs y^* . This would make the presentation of results consistent with those in figure 2a and would more clearly show if the model is correctly downshifting or upshifting the parameters if they are over- or under-predicted. Perhaps this is inconvenient though if there is some error cancellation of κ and B that is currently being masked in 2b.

Comments on the section "Turbulent channel flow"

- The name of this section is confusing since I thought the last section on maps was also about channels. Perhaps it just needs to be more specific, for example, "Testing higher Reynolds number channels."
- There is a claim that the LLWM is superior to the classical EQWM because the latter uses an "empirical coefficient tuned for this particular case." This is a bit misleading since the EQWM uses coefficients that are generally not tuned on a case-by-case basis, even if those coefficients were originally tuned using channel data. I would say that present approach also has some assumed truth values for the coefficients (which I am unsure how these are determined). If the EQWM fails because the best choices of its model coefficients differ from the assumed values, I'm not sure if LLWM will

outperform unless its assumed truth values are adapted. Such automatic adaptation is not demonstrated in the paper, but somehow seem to be claimed throughout. If manual adaptation is required for the LLWM, please note that manual adaptation is also possible for the EQWM. It would be helpful if the authors show a case where the truth values automatically adapt, like in a pressure gradient boundary layer where κ and B are different than their default values.

Figure 3

- The dotted horizontal line does not go through zero. What is this line?
- It is difficult to distinguish much of the data and impossible to discern the magnitude of the error. Please replot with log scale. Or show a separate figure with the VWM data omitted.
- I can see that the error for LLWM is growing with Re_{τ} . There should be a remark in the text that acknowledges this. This is important since the central claim of this paper is that the LLWM is learning κ and B and no DNS simulations are required to learn this information. However, it appears at high Re_{τ} , that the procedure is not predicting the non-universality (Reynolds number dependence) of κ and B that has been reported widely in the literature. Why should LLWM perform better at low Re_{τ} , if not for an issue with the truth values of κ and B .
- Please compare against the accuracy of the EQWM. Throughout the paper the EQWM is regarded as the state of the art, but it is not compared against in this key plot, so it is impossible for the reader to compare the present approach with the state of the art in this canonical setting.

Comments on the section "Potential of SciMARL wall models"

- Regarding the limitations of the EQWM, I have a similar comment to the one that I made in the section "Turbulent channel flow." It is unclear to me that their model is less limiting than the EQWM.
- "do not require any DNS simulation data." the accuracy of this statement depends on how the authors address my comment regarding truth values κ , B , and τ_w .
- The comparison of LLWM with supervised learning may not be appropriate (depending on how the authors defined their ground truth values as questioned above). If the truth values for κ and B are from DNS/experiments, then the claimed advantage of RL is not present.
- "single model trained from various canonical roughness cases." It is not clear to me how the authors can generalize this model to rough walls without providing some roughness characteristics to the model and also modified truth values. Please provide additional information or equations.

Comments on the section "Discussion"

- The phrase in the discussion section "present a paradigm shift" is perhaps an overly strong claim. The present approach has not been shown to be more predictive than existing methods and hasn't been tested in complex cases that testing methods have succeeded in. The usefulness of the model has been speculated, but not demonstrated yet.

Comments on the section "Methods- Reinforcement learning":

- Please explicitly define the term "episode."
- Please provide a reference of clipping "far-policy" experiences as this seems ad hoc. "One way ..." it is unclear if the authors are proposing this strategy or if it has been used previously.

Comments on the section "Methods- Overview of the training setup":

- Only one resolution is used in x and z . Demonstrating other resolutions would be useful for establishing robustness. Especially since the boundary layer also has very similar resolution.
- Change "uniformly sampled" to "spatially uniformly sampled"
- Robustness to numerics is claimed, but this is not tested.
- Why is white noise added to the training data set. Is this to create additional realizations? If so, I don't see how these additional realizations would be more informative than a single realization. Please explain.
- How did the authors decide on their network topology. Do they have any references or did they conduct a study on the sensitivity to the network architecture?

Comments on the sections called "Testing"

- Please clarify this sensitivity comment.
- Perhaps here the authors should cite Kawai and Larsson for using the third point matching location: "third grid point off the wall in the wall-normal direction"

Response to Referee #1

H. J. Bae & P. Koumoutsakos

October 8, 2021

We are grateful for the Referees' insightful comments and suggestions for improving the paper. In the revised manuscript we have addressed all the comments of the referee and made modifications to further clarify the value and originality of this contribution. The relevant modifications in response to the comments of the Referee have been made in blue.

1. *Please provide more comments about the advantages of SciMARL models, as well as the dependency of SciMARL models on the training data or the predefined theoretical model. In the situation of the velocity-based wall model (VWM), the training processes are dependent on the LES of a turbulent channel flow at moderate Reynolds numbers, and the model cannot be easily generalized to turbulent channel flow at very high Reynolds numbers. The limitation of the proposed VWM is similar to the situations of supervised learning models, which are limited by the scope of training data. In the situation of the log-law-based wall model (LLWM), the training processes are dependent on the well-known theoretical log-law model. For the general complex flows which are not consistent with the theoretical log-law model, the proposed LLWM could fail. For the flows which are consistent with the theoretical log-law model, the original log-law model is enough.*

We have followed the suggestion of the Referee, and in the revised manuscript we provide further information on the SciMARL models, their data dependency, and their training. The referee is correct about the advantages and limitations of the VWM. The LLWM is based on the existence of a log law, which can be theoretically shown to exist for any flow that has a scale separation in the inner and outer region (Millikan 1939). However, we wish to clarify that the LLWM does not take as input the log-law parameters and we have added to the text, the following sentence: “We emphasize that this model does not take as input the *a priori* known values of κ and B from the log-law, but rather derived quantities from the instantaneous flow” by calculating $y^* \partial u^* / \partial y^*$ and $u^* - y^* \partial^* / \partial y^* \log y^*$ in the current flow field.

In addition, any flow with a solid boundary will indeed exhibit a log-law, although with different values of κ and B . We now state that “These flows [flows exhibiting roughness, stratified flows, compressible flows, among many others] usually have different log-law coefficients κ and B that are manually tuned for existing wall models. However, in the present work, these values are adjusted automatically using a SciMARL-based model.”

We note that this is a key advantage of SciMARL-based models over models such as EQWM, as they can adapt to various flow configurations without manual changing of the coefficients.

We also state in the manuscript that “Furthermore, the RL framework can be extended to various flow configurations by adding an additional dimension to the state vector. Since all flow with an inner-outer scale separation exhibit a log law in the overlap region, the current configuration for wall-model development can be extended to flows exhibiting roughness, stratified flows, compressible flows, among many others... This gives the LLWM a distinct advantage over existing models. For example, in cases with varying roughness over the simulation domain, traditional methods will have to assign different model parameters for each patch containing different roughness heights. In contrast, the SciMARL model can smoothly transition between various roughness elements with a single policy trained from various canonical roughness cases when the roughness height is included as a state”. As such, additional flow features can be addressed by augmenting the state vector with quantities that best reflect the influence of this feature on the flow field. We hope to address this in a future study, as the present paper represents a proof of concept of the methodology, and such extensions of the model will require extensive simulations.

2. *Please provide the computational cost for LES of wall-bounded turbulent flows using the new models, and make a comparison with traditional wall models.*

We now include a sentence in the manuscript stating that “the evaluation of the LLWM involves evaluating the weights of the trained neural net, which is an order of magnitude faster than the EQWM that solves an ODE at each time step.”

3. *Please give some comments about the effects of different parameters of SciMARL models on the accuracy of LES.*

We now include comments regarding the different parameters of SciMARL in the Methods section:

“The most notable hyper-parameters used in our description of the MARL set-up are the spatial resolution for the interpolation of the actions onto the grid (determined by Δ_x^m/Δ_x , and Δ_z^m/Δ_z). The default values Δ_x^m/Δ_x , and Δ_z^m/Δ_z reduce the number of experiences generated per simulation to $O(10^5)$. This value is similar to the number of experiences generated per simulation used for SciMARL of SGS model development. Consistent with previous studies, we found that further reducing the number of agents per simulation reduced the model’s adaptability and therefore exhibit slightly lower performance. Because we use conventional reinforcement learning update rules in a multi-agent setting, single parameter updates are imprecise. We found that ReF-ER with hyper-parameters $C = 1.5$ and $D = 0.05$ (Eqs. (6) and (7)) stabilizes training. We ran multiple training runs per reward function and whenever we vary the hyper-parameters, but we observe consistent training progress regardless of the initial random seed.”

Response to Referee #2

H. J. Bae & P. Koumoutsakos

October 8, 2021

We are grateful for the Referees' critical feedback and detailed suggestions for improving the paper. In the revised manuscript we have addressed all the comments of the referee and made modifications to further clarify the value and originality of this contribution. The relevant modifications in response to the comments of the Referee have been made in **red**.

1. *General comments*

- *The authors have claimed that their approach is a significant advance over the state of the art because rather than assume κ and B , it has learned them through unsupervised learning. However, the manuscript does not state how the truth values of these turbulence coefficients (κ and B) are established and provided to the reinforcement learning agents.I have given the authors the benefit of the doubt and tried to write this review under the assumption that the authors can modify the manuscript to demonstrate that κ and B are discovered rather than assumed.*

We appreciate the comments of the Referee and we apologize that in the previous version of the manuscript it was not clear that **the values of κ and B are indeed discovered by the multi-agent reinforcement learning**. In the following, we further stress this point as it is a key contribution of the paper.

2. *Comments on the introduction*

- *“such wall models can only provide limited success” It might be helpful to add references that outline some of the regimes where the RANS models are failing (separated flows, predicting transition, etc.) to help motivate the present work. “Limited success” is fairly vague.*

Following the suggestion of the Referee, we now specify that “Such wall models do not function as intended in real-world applications, where various flow states coexist (e.g. separated flows, flow over roughness, predicting transition, etc.)” and we provide additional references.

- *“SciMARL does not rely on a priori knowledge” the accuracy of this statement depends on how the authors address my comment regarding truth values κ , B , and τ_w .*

The SciMARL-based wall models do not need an explicit input of the log-law coefficients κ and B , which is necessary for the traditional RANS-based models.

Instead, it only requires the mean wall-shear stress for a few Reynolds numbers, which can be readily obtained from existing data sets. In canonical flow over flat-plates, the values of κ and B are widely studied and considered constants. However, this is not generalizable for flows configurations with roughness, pressure-gradient effects, or compressibility effects. While manually setting the RANS-based wall models can be done, the ultimate purpose of WMLES is to provide predictions where the features of the flow configuration are not known *a priori*.

In the current paper, we show the potential of SciMARL-based wall models to perform as well as RANS-based wall models in canonical zero-pressure-gradient flat-plate boundary layer flow. We also argue that the SciMARL-based wall models “can be extended to various flow configurations [(e.g. roughness, pressure-gradient effects, compressibility effects)] by adding an additional dimension to the state vector.” This will allow the SciMARL wall models to adapt naturally to the given flow configuration, unlike RANS-based models that need to be manually tuned.

3. Comments on the section “Results”

- *The “results” section also describes the model. Is this common in this journal?*

The journal allows only three sections (Introduction, Results, and Discussion) in the main body of the paper. The separate Methods section, which comes after the References, is supposed to include the methodology used to produce the results. We opted to introduce the models themselves in the Results section in order to explain their training within the main text of the article.

- *Eq. 1, the notation for the expectation under policy π should be explicitly mentioned since presently \mathbb{E}_π is not defined. The summation over n is not defined. n is defined in the methods section, but must be defined when it is used in Eq. 1. It would be helpful to note that γ will be specified in the methods section since it was unclear that this was a constant parameter when it was introduced in the results section and not specified.*

We regret the confusion introduced by this notation. To maintain consistency, with the rest of the manuscript we removed Eq. 1 from the main text. We now write that: “The optimal policy $\pi^*(s, a)$ is found by maximizing the expected utility, which is given by the expected cumulative reward.”

- *A single policy is made by all the agents. This seems like it is only the case in homogeneous flows. In general problems, there will need to be n agents with n policies. In a channel, it seems like only 1 agent is needed. In a boundary layer, one would need more than 1 agent if κ and B are thought to vary with the streamwise coordinate. (However, this variation is likely modest.) Are the authors using additional agents in the span just to improve the statistical convergence?*

We agree with the Referee that using multiple agents allows for better statistics and in turn a robust, collective formulation of the agents’ policy that is also adjusted to their position in the flow field.

We note that even in non-homogeneous flows (e.g. flow with varying pressure gradients or roughness height), a single policy can be maintained by all the agents.

The state-space will include additional information in this case (such as pressure gradient), which will allow the agents to determine the correct policy based on those additional inputs. This is indeed the primary benefit of SciMARL, which allows automatic adjustment of the log-law coefficients based on flow configuration. We now include an example to highlight this feature: “... the SciMARL model can smoothly transition between various pressure-gradient effects with a single policy trained from various canonical cases when the parameters such as pressure and velocity gradients are included as a state”.

In the channel case, multiple agents are necessary as the agents themselves act as the wall model. This is equivalent to solving the ODE associated with the EQWM at all wall points in the computational domain.

4. *Comments on the section “Velocity-based wall model”*

- *It might be helpful to refer the reader to the methodology section since the term “reward” is used without being defined. Especially the bonus to the reward that is designed to promote stability. The reader may not realize that this will be defined later.*

We agree and we now refer the reader to the methods section.

5. *Comments on the section “State-action map”*

- *I don’t see where the authors mention which flow they are plotting. I assume it is a channel.*

The Referee is correct. We now include a text that mentions that the results are “for the channel flow at friction Reynolds number $Re_\tau = 2000, 4200, 8000$ ”.

- *For figure 2, I think the a simpler explanation will be possible by splitting 2b into 2 panels (2b and 2c) for plots of κ vs y^* and B vs y^* . This would make the presentation of results consistent with those in figure 2a and would more clearly show if the model is correctly downshifting or upshifting the parameters if they are over- or under-predicted. Perhaps this is inconvenient though if there is some error cancellation of κ and B that is currently being masked in 2b.*

The main goal of figure 2 was to show the state and action map of the two trained wall models, VWM and LLWM. Since the states for the LLWM are $1/\kappa$ and B , we decided that the current form on figure 2b was the most appropriate.

6. *Comments on the section “Turbulent channel flow”*

- *The name of this section is confusing since I thought the last section on maps was also about channels. Perhaps it just needs to be more specific, for example, “Testing higher Reynolds number channels.”*

We have changed the subsection title to “Testing: channel flow”. We also changed the following subsection to “Testing: Spatially evolving turbulent boundary layer”

- *There is a claim that the LLWM is superior to the classical EQWM because the latter uses an “empirical coefficient tuned for this particular case.” This is a bit misleading since the EQWM uses coefficients that are generally not tuned on a case-by-case basis, even if those coefficients were originally tuned using channel data. I would say that the present approach also has some assumed truth values for the coefficients (which I am unsure how these are determined). If the EQWM fails because the best choices of its model coefficients differ from the assumed values, I’m not sure if LLWM will outperform unless its assumed truth values are adapted. Such automatic adaptation is not demonstrated in the paper, but somehow seem to be claimed throughout. If manual adaptation is required for the LLWM, please note that manual adaptation is also possible for the EQWM. It would be helpful if the authors show a case where the truth values automatically adapt, like in a pressure gradient boundary layer where κ and B are different than their default values.*

We emphasize that the proposed sciMARL models do not use as input known values of κ or B . In fact ‘estimates’ of κ and B obtained by calculating $y^* \partial u^* / \partial y^*$ and $u^* - y^* \partial^* / \partial y^* \log y^*$ in the current flow field are used as states for the agents. These states are changing over time and they are used to inform the agent on what action to take in order to maximize its reward.

On the contrary, EQWM requires the value of κ in the model as well as the damping coefficient, which indirectly relates to the value of B . We mention “empirical coefficient tuned for this particular case” to indicate that the EQWM requires the κ and B as parameters of the model. While the training of RL models requires the true value of τ_w at low Reynolds number cases, τ_w is a known quantity obtained from the imposed pressure gradient for channel cases.

Additional flow features can be addressed by augmenting the state vector with quantities that best reflect the influence of this feature on the flow field. We hope to address this in a future study, as the present paper represents a proof of concept of the methodology, and such extensions of the model will require extensive simulations.

7. Figure 3

- *The dotted horizontal line does not go through zero. What is this line?*

Thank you for catching this mistake. The line was drawn at 1% error by mistake. The dotted line is meant to be zero error and is now located at zero.

- *It is difficult to distinguish much of the data and impossible to discern the magnitude of the error. Please replot with log scale. Or show a separate figure with the VWM data omitted.*

Due to the negative error, we cannot plot the error in log scale; however, we now include a zoom-in of the figure from -1 to 5% in figure 3(b). We now also include the EQWM results in the zoomed-in figure for comparison.

- *I can see that the error for LLWM is growing with Re_τ . There should be a remark*

in the text that acknowledges this. This is important since the central claim of this paper is that the LLWM is learning κ and B and no DNS simulations are required to learn this information. However, it appears at high Re_τ , that the procedure is not predicting the non-universality (Reynolds number dependence) of κ and B that has been reported widely in the literature. Why should LLWM perform better at low Re_τ , if not for an issue with the truth values of κ and B .

We agree with the Referee and we now include a short discussion on this trend: “The error increases with Reynolds number, most likely due to the high variation of the streamwise wall-normal gradient with increasing Reynolds number as well as the departure of $(h^m)^+$ from the trained range of values.”

- *Please compare against the accuracy of the EQWM. Throughout the paper the EQWM is regarded as the state of the art, but it is not compared against in this key plot, so it is impossible for the reader to compare the present approach with the state of the art in this canonical setting.*

We now include the errors of the EQWM for the same Reynolds number and grid resolutions. While the errors for EQWM are smaller than those of LLWM, this is expected since EQWM has information regarding the true log-law embedded in the model itself. We now mention that “Still, the results are comparable to the results obtained from the widely-used equilibrium wall model (EQWM) up to $Re_\tau \approx 10^5$, which uses an empirical coefficient tuned for this particular flow configuration. This range of Reynolds numbers is sufficient for various external aerodynamic and geophysical flows.”

8. Comments on the section “Potential of SciMARL wall models”

- *Regarding the limitations of the EQWM, I have a similar comment to the one that I made in the section “Turbulent channel flow.” It is unclear to me that their model is less limiting than the EQWM.*

We hope the previous response also addresses this comment and it is satisfactory for the Referee.

- *“do not require any DNS simulation data.” the accuracy of this statement depends on how the authors address my comment regarding truth values κ , B , and τ_w .*

We hope the previous response is satisfactory for the referee.

- *The comparison of LLWM with supervised learning may not be appropriate (depending on how the authors defined their ground truth values as questioned above). If the truth values for κ and B are from DNS/experiments, then the claimed advantage of RL is not present.*

We remark that the truth values of κ and B are not used in the RL model. Supervised learning requires (a lot of) instantaneous snapshots that are not readily available publicly. To obtain this data, either a new DNS needs to be run to save the instantaneous flow fields or access to prior databases that store such data is necessary. On the contrary, finding averaged quantities for τ_w is much easier and readily available in the literature. Additionally, the storage cost alone is advantageous for RL models.

- *“single model trained from various canonical roughness cases.” It is not clear to me how the authors can generalize this model to rough walls without providing some roughness characteristics to the model and also modified truth values. Please provide additional information or equations.*

We appreciate this observation of the Referee. Indeed in order to address flows with features such as roughness the training of the RL should include flows that exhibit such features. Additional flow features can be addressed by augmenting the state vector with quantities that best reflect the influence of this feature on the flow field. We hope to address this in a future study, as the present paper represents a proof of concept of the methodology, and such extensions of the model will require extensive simulations.

9. Comments on the section “Discussion”

- *The phrase in the discussion section “present a paradigm shift” is perhaps an overly strong claim. The present approach has not been shown to be more predictive than existing methods and hasn’t been tested in complex cases that testing methods have succeeded in. The usefulness of the model has been speculated, but not demonstrated yet.*

We respectfully note that the word “paradigm shift” refers to the way the wall model is derived. In classical wall modeling, there is an *a priori* specification of the form of the wall model and the parameters are obtained by comparing with existing results. The present method allows for on-the-fly adaptation of the wall model using current information. Moreover, while past methodologies first obtain the model and then apply it, here the model is adapted while being applied. We believe that this is indeed a “paradigm shift” and unique to the SciMARL methodology that we have introduced. The Referee is correct in observing that we have not demonstrated that our approach is superior to everything else, but we believe that the methodology is new and can lead to major advances by ourselves and others that would adopt and extend this approach.

In the revised manuscript we now specify that “paradigm shift” refers to the methodology and stress the two points mentioned above: “This advance will present a paradigm shift in wall model development for LES in the prediction and control of industrial aerodynamics and environmental flows.”

10. Comments on the section “Methods- Reinforcement learning”:

- *Please explicitly define the term “episode.”*

We mention in the text “The flow simulations are distributed across workers who collect, for each agent, experiences organized into episodes,

$$E_i = \{s_n^{(i)}, r_n^{(i)}, \mu_n^{(i)}, \sigma_n^{(i)}, a_n^{(i)}\}_{n=0, \dots, N},$$

where n tracks in-episode RL steps, μ and σ are the statistics of the Gaussian policy used to sample a , and t_N is the final time step for each episode.”

- Please provide a reference of clipping “far-policy” experiences as this seems ad hoc. “One way ...” it is unclear if the authors are proposing this strategy or if it has been used previously.

We now state that “We circumvent this issue by...” to make sure to convey that this is how we make sure that the policy does not diverge from the distribution of experiences in the RM. We cite Ref. 52 as the reference for clipping far-policy experiences. More details can be found in the publication.

11. Comments on the section “Methods- Overview of the training setup”:

- Only one resolution is used in x and z . Demonstrating other resolutions would be useful for establishing robustness. Especially since the boundary layer also has very similar resolution.

Figure S1: Mean velocity profile of channel flow at $Re_\tau = 10^5$ using LLWM. The resolutions in the streamwise and spanwise directions are $\Delta x \approx \Delta z \approx 0.05\delta$ (circle), $\Delta x \approx 0.1\delta$ and $\Delta_z \approx 0.05\delta$ (cross), and $\Delta x \approx 0.05\delta$ and $\Delta_z \approx 0.1\delta$ (triangle).

We have conducted two simulations, using the LLWM, where the resolution in x and z , respectively, are coarsened by a factor of 2. The mean velocity profile for $Re_\tau = 10^5$ is shown for the original resolution and the two coarsened cases in figure S1. The difference is almost nonexistent, and any difference should be due to the SGS model rather than the wall model. This is because the wall model takes only the information at $y = h_{wm}$. If the SGS model is capable of providing the correct eddy viscosity, the wall model will produce similar results regardless of resolution. Due to the limitation of the journal format (limit on the total number of figures), we cannot include an additional figure, but we add in the manuscript (Testing: Channel flow) that “While only results using $\Delta_x \approx \Delta_z \approx 0.05\delta$ are reported here, using different grid resolutions representative of WMLES also produce similar results.”

- Change “uniformly sampled” to “spatially uniformly sampled”

We use “uniformly sampled” to indicate that the Re_τ for the simulation is chosen at random, with equal probability, from the set $\{2000, 4200, 8000\}$. We feel that “spatially uniformly sampled” is not accurate for this description.

- *Robustness to numerics is claimed, but this is not tested.*

There are two claims to robustness. One is robustness to *a posteriori* testing as opposed to *a priori* testing of models. This is a big caveat to supervised learning, where the model may perform well for one time-step predictions, but fails to perform well when the errors accumulate in time. Hence, we claim that “The [SciMARL] wall models are robust with respect to the numerical discretizations, as these errors are taken into consideration in the training process” in the introduction. This is a key finding in reinforcement learning in general.

The second is that we “expect” robustness to different numerics as only velocity and velocity gradient information away from the wall is required. We mention in the text “The LLWM is easy to extend to complex geometries and flow simulations utilizing different numerical methods or SGS models, as it only takes as states the instantaneous streamwise (or wall-parallel) velocity, its wall-normal gradient, and the distance from the wall. These quantities do not depend heavily on numerics or SGS models, unlike filtered velocities or eddy viscosity values required in the dynamic model.” This is also the reason that wall models such as EQWM are robust to numerics. We feel that this does not overstate our findings with the current model.

- *Why is white noise added to the training data set. Is this to create additional realizations? If so, I don't see how these additional realizations would be more informative than a single realization. Please explain.*

The Referee is correct—the additional white noise is to create additional realizations. In particular, this was to avoid the learning process from overtraining on a particular initial condition to the point the actions become deterministic.

- *How did the authors decide on their network topology. Do they have any references or did they conduct a study on the sensitivity to the network architecture?*

We now include comments regarding the different parameters of SciMARL models in the Methods section:

“The most notable hyper-parameters used in our description of the MARL set-up are the spatial resolution for the interpolation of the actions onto the grid (determined by Δ_x^m/Δ_x , and Δ_z^m/Δ_z). The default values Δ_x^m/Δ_x , and Δ_z^m/Δ_z reduce the number of experiences generated per simulation to $O(10^5)$. This value is similar to the number of experiences generated per simulation used for SciMARL of SGS model development. Consistent with previous studies, we found that further reducing the number of agents per simulation reduced the model’s adaptability and therefore exhibit slightly lower performance. Because we use conventional reinforcement learning update rules in a multi-agent setting, single parameter updates are imprecise. We found that ReF-ER with hyper-parameters $C = 1.5$ and $D = 0.05$ (Eqs. (6) and (7)) stabilizes training. We ran multiple training runs per reward function and whenever we vary the hyper-parameters, but we observe consistent training progress regardless of the initial random seed.”

12. *Comments on the sections called “Testing”*

- *Please clarify this sensitivity comment.*

We have changed the manuscript to: “For example, if y is located at the midpoint of two computational grid points, a central finite difference can be used to compute the wall-normal derivative $\partial u^*/\partial y^*$. On the other hand, if y is located on the computational grid point, either a left- or right- finite difference is used. In this case, we chose y values that are midpoints of the two computational grid points. Changing the location of y had minor effects on the results, with the wall-shear stress changing $\sim 5\%$ when the location of y was chosen to be on the computational grid point.” We hope the text is more clear now.

- *Perhaps here the authors should cite Kawai and Larsson for using the third point matching location: “third grid point off the wall in the wall-normal direction”*

We now cite Kawai and Larsson for the use of the third grid point off the wall.

REVIEWER COMMENTS

Reviewer #2 (Remarks to the Author):

I appreciate that the authors have addressed most of my comments, but I believe that the novelty and utility of using reinforcement learning for this problem is overstated. More specifically, the revised manuscript does not provide a convincing argument why SciMARL should be favored over supervised learning or the classical wall modeling approach. Although the authors have acknowledged this criticism, I believe that this key question remains unresolved.

1. This comment is postponed for address in subsequent remarks.

2. I maintain that the author's statement "SciMARL does not rely on a priori knowledge" is misleading. Training the model relies on knowledge of the exact wall shear stress. Even in a canonical boundary layer, this requires a high fidelity simulation (DNS) as a priori knowledge that is used to define the reward for the SciMARL method. The quotation and similar statements throughout the manuscript should be removed. Surrounding discussions should be revised to not claim priority of SciMARL over other methods that also require a priori knowledge.

3. This comment has been resolved.

4. This comment has been resolved.

5. This comment has been resolved.

6. I don't accept the author's claims of the advantages of SciMARL over the classical EQWM. By providing the exact shear stress data in the training of SciMARL, the method learns the appropriate values of κ and B . Similarly, the classical EQWM is provided with empirical values of κ and B . So both rely on empirical information that is case specific (insofar as κ and B vary in non-canonical cases). The EQWM and SciMARL will both need to be retrained in flows where the optimal values of κ and B vary.

a. The authors correctly comment that SciMARL is a framework that could be provided with the roughness or pressure gradient to be applied to complex flows. Meanwhile, the law of the wall (or EQWM) has already been extended to depend on roughness parameters (see the review by J. Jimenez "Turbulent flows over rough walls" ARFM). The authors should acknowledge that both SciMARL and the EQWM depend on empirical training data and both can be retrained in complex settings by supplying additional data.

b. In the rebuttal, the authors argue that the exact shear stress is not external information since the channel is driven by a constant pressure gradient, so the exact shear stress is available from the pressure gradient.

i. Firstly, this is only true in a channel, so this argument does not generalize to other flows, so my criticism stands in general flows.

ii. Secondly, I'm not convinced that the channel is actually run with an applied constant pressure gradient. In such a setting, the mean wall stress is determined by the applied pressure gradient, so mean wall stress will be correct regardless of the wall model (by force balance). How then, can providing SciMARL with the exact wall stress be any different than providing it with the mean modeled stress? By this argument, the reward (equation 8) incentivizes a policy that minimizes wall stress fluctuations. What is the physical motivation for such a reward?

7. This comment has been resolved.

8. The authors are comparing their SciMARL approach to a particularly unattractive flavor of supervised learning. They assume that supervised learning will involve storing many instantaneous flow fields to train the model, and thus requires large amounts of data storage. This black-box approach of using the entire instantaneous flow field will probably not be as successful as an alternative approach that assumes the existence of a log law (similar to the authors' SciMARL approach or others in the literature). After making this assumption, the supervised learning approach only depends on the mean velocity profile, which is ubiquitously reported in the literature, and has negligible storage requirements. Using such a profile and assuming a log law, one can fit κ and B , which is the type of supervised learning that has been traditionally used for this problem (the classical log law). In a more complicated setting, the pressure gradient and wall roughness can also be provided as inputs and a neural network can be used to provide a more complex model for κ and

B. I don't see how the author's method is less costly or more accurate than this physics-informed supervised learning approach. I think it is a false comparison to compare their SciMARL approach to a physics-free black-box approach that uses the whole velocity field and does not make the log law assumption.

9. This comment has been resolved.

10. This comment has been resolved.

11. This comment has been resolved.

12. This comment has been resolved.

Response to Referee #2

H. J. Bae & P. Koumoutsakos

December 13, 2021

I appreciate that the authors have addressed most of my comments, but I believe that the novelty and utility of using reinforcement learning for this problem is overstated. More specifically, the revised manuscript does not provide a convincing argument why SciMARL should be favored over supervised learning or the classical wall modeling approach. Although the authors have acknowledged this criticism, I believe that this key question remains unresolved.

We are grateful for the Referee’s critical feedback and detailed suggestions for improving the paper. In the revised manuscript we have addressed all the comments of the Referee and made modifications to further clarify the value and originality of this contribution. We respectfully maintain that the present method provides a major contribution to the development of wall models for turbulent flows. Moreover, by addressing this challenging problem, we believe that we provide evidence for a novel, and in our opinion potent, way of combining numerical methods and learning algorithms to develop closures for under-resolved equations of complex systems.

In the following, we list the comments of the Referee in **red** (*using the numbering of the Referee bullets*) followed by our answers. The relevant modifications in the manuscript are also noted in **red**.

2. I maintain that the author’s statement “SciMARL does not rely on a priori knowledge” is misleading. Training the model relies on knowledge of the exact wall shear stress. Even in a canonical boundary layer, this requires a high fidelity simulation (DNS) as a-priori knowledge that is used to define the reward for the SciMARL method. The quotation and similar statements throughout the manuscript should be removed. Surrounding discussions should be revised to not claim priority of SciMARL over other methods that also require a-priori knowledge.

The Referee is right that RL relies on a-priori knowledge. However we wish to emphasize the very limited amount of information that is necessary for its training. For example sciMARL requires only global state information (such as total drag) and not detailed spatio-temporal information. The limited data is a key feature of SciMARL as a semi-supervised learning algorithm in contrast to supervised learning algorithms.

We have modified the text to “SciMARL is a semi-supervised learning algorithm that requires information about the flow formulated in terms of a reward rather than detailed spatiotemporal data as in the case of supervised learning. In the case of wall

modeling, we emphasize that SciMARL does not rely on *a priori* knowledge of the log-law coefficients but rather aims to discover active closure policies according to patterns in the flow physics captured by the filtered equations.”

6. I don't accept the author's claims of the advantages of SciMARL over the classical EQWM. By providing the exact shear stress data in the training of SciMARL, the method learns the appropriate values of κ and B . Similarly, the classical EQWM is provided with empirical values of κ and B . So both rely on empirical information that is case specific (insofar as κ and B vary in non-canonical cases). The EQWM and SciMARL will both need to be retrained in flows where the optimal values of κ and B vary.

We respectfully disagree. In fact the comment of the Referee points exactly to the difference between SciMARL and EQWM: **SciMARL learns effective representations of the κ and B coefficients while the EQWM is given the values of the κ and B coefficients.**

In addition, the SciMARL model neither knows a-priori nor learns the coefficient directly. Instead SciMARL learns how to control the wall shear stress based on the instantaneous velocity and velocity gradient measurements from the LES itself. This will be beneficial in cases such as adverse pressure gradient flow, where the resulting adverse pressure gradient due to the geometry is not known *a priori*. Thus, the model cannot be tuned in advance to capture the correct flow—the SciMARL model would be capable of adapting within the simulation, while the EQWM cannot.

Moreover, obtaining the κ and B for non-canonical flows is not a straightforward process, whereas obtaining the mean wall-shear stress for a particular case is an easier task. The consensus on the values of κ and B for the flat plate case was only reached after decades of high Reynolds number experiments on this geometry. For adverse-pressure-gradient cases, this will require high-Reynolds number simulations and experiments over various values of adverse-pressure gradients, which may not be feasible in the near future.

- a. The authors correctly comment that SciMARL is a framework that could be provided with the roughness or pressure gradient to be applied to complex flows. Meanwhile, the law of the wall (or EQWM) has already been extended to depend on roughness parameters (see the review by J. Jimenez “Turbulent flows over rough walls” ARFM). The authors should acknowledge that both SciMARL and the EQWM depend on empirical training data and both can be retrained in complex settings by supplying additional data.

We agree with the Referee that SciMARL requires appropriate training in order to generalise to various flows. In fact SciMARK requires physical insight to facilitate its training and does not promise to work without having physical knowledge about turbulent flows. In order to extend SciMARL to other flows physical insight is necessary for the appropriate choice of states and actions that SciMARL entails. In that sense SciMARL provides a new and more flexible set of tools for developing

wall models when compared to the EQWM. We iterate that EQWM requires a-priori knowledge of the appropriate κ and B while SciMARL does not. However, here we do not argue that EQWM will not be successfully formulated and extended to various flows. Indeed there are several excellent researchers that are working in this directions as discussed in the ARFM article of Jimenez. What we propose is an alternative to the EQWM.

- b. In the rebuttal, the authors argue that the exact shear stress is not external information since the channel is driven by a constant pressure gradient, so the exact shear stress is available from the pressure gradient.
 - i. Firstly, this is only true in a channel, so this argument does not generalize to other flows, so my criticism stands in general flows.
 - ii. Secondly, I'm not convinced that the channel is actually run with an applied constant pressure gradient. In such a setting, the mean wall stress is determined by the applied pressure gradient, so mean wall stress will be correct regardless of the wall model (by force balance). How then, can providing SciMARL with the exact wall stress be any different than providing it with the mean modeled stress? By this argument, the reward (equation 8) incentivizes a policy that minimizes wall stress fluctuations. What is the physical motivation for such a reward?

The Referee is correct. We made an error in the previous rebuttal regarding how the channel flow is driven. The training process was indeed run using a constant mass flow rate, which was obtained from the mean velocity profile of the channel flow. The mean velocity profile for these case is readily available in the literature for low to moderate Reynolds numbers for a wide range of cases (including pressure gradient effects), which is all that is necessary for the training of SciMARL. This provides information for both the mean wall-shear stress and the mass flow rate.

The testing process was run with a constant pressure gradient, as we indicated before in our previous response.

We hope that the above statements help clarify an issue that appears to be at the core of several questions of the Referee.

- 8. The authors are comparing their SciMARL approach to a particularly unattractive flavor of supervised learning. They assume that supervised learning will involve storing many instantaneous flow fields to train the model, and thus requires large amounts of data storage. This black-box approach of using the entire instantaneous flow field will probably not be as successful as an alternative approach that assumes the existence of a log law (similar to the authors' SciMARL approach or others in the literature). After making this assumption, the supervised learning approach only depends on the mean velocity profile, which is ubiquitously reported in the literature, and has negligible storage requirements. Using such a profile and assuming a log law, one can fit κ and B , which is the type of supervised learning that has been traditionally used for this problem (the classical log law). In a more complicated setting, the pressure gradient

and wall roughness can also be provided as inputs and a neural network can be used to provide a more complex model for κ and B . I don't see how the author's method is less costly or more accurate than this physics-informed supervised learning approach. I think it is a false comparison to compare their SciMARL approach to a physics-free black-box approach that uses the whole velocity field and does not make the log law assumption.

We report on published supervised learning approaches that use many instantaneous flow fields to develop data for training their models. We do not disagree that what the Referee suggest may be a viable alternative and we will be happy to provide references if they are available. At the same time we wish to iterate that even in that case, supervised learning will require information about the parameters of the law of the wall. On the other hand, the wall model discovered through SciMARL, by including additional states, can adapt to flow conditions without specifying what the log law for that particular flow should be. The EQWM and the suggested supervised learning of the EQWM behavior still need this external information (for example, what type of log-law to follow at a particular location) and need to be re-calibrated based on the flow, which may not be feasible in complex settings.

REVIEWERS' COMMENTS

Reviewer #2 (Remarks to the Author):

Thank you for the detailed response. All of my comments have been addressed. I believe that the revised manuscript clearly represents its novel contribution to this important field of study and that the methods and results are clearly and accurately presented and interpreted.